# ATTENTION: SELF-EXPRESSION IS ALL YOU NEED

## ABSTRACT

Transformer models have achieved significant improvements in performance for various learning tasks in natural language processing and computer vision. Much of their success is attributed to the use of attention layers that capture long-range interactions among data tokens (such as words and image patches) via attention coefficients that are global and adapted to the input data at test time. In this paper we study the principles behind attention and its connections with prior art. Specifically, we show that attention builds upon a long history of prior work on manifold learning and image processing, including methods such as kernel-based regression, non-local means, locally linear embedding, subspace clustering and sparse coding. Notably, we show that self-attention is closely related to the notion of self-expressiveness in subspace clustering, wherein data points to be clustered are expressed as linear combinations of all other points with coefficients designed to attend to other points in the same group, thus capturing long-range interactions. We also show that heuristics in sparse self-attention can be studied in a more principled manner using prior literature on sparse coding and sparse subspace clustering. We thus conclude that the key innovations of attention mechanisms relative to prior art are the use of many learnable parameters, and multiple heads and layers.

## 1 INTRODUCTION

Attention, i.e., the ability to selectively focus on a subset of sensory observations, while ignoring other irrelevant information, is a central component of human perception. For example, only a few words in a sentence may be useful for predicting the next word, or only a small portion of an image may be relevant for recognizing an object. This property of biological systems has inspired the recent development of attention-based neural architectures (Bahdanau et al., 2014), such as Transformers (Vaswani et al., 2017), BERT (Devlin et al., 2018), GPT (Radford et al., 2018; 2019), RoBERTa (Liu et al., 2019), and T5 (Raffel et al., 2019), which have achieved impressive performance in multiple natural language processing tasks, including text classification (Chaudhari et al., 2019; Galassi et al., 2020), machine translation (Ott et al., 2018), and question answering (Garg et al., 2020). Attention-based architectures have also led to state-of-the-art results in various computer vision tasks (Khan et al., 2021), including image classification (Dosovitskiy et al., 2020), object detection (Carion et al., 2020; Zhu et al., 2020), and visual question answering (Tan & Bansal, 2019; Su et al., 2019).

Much of the success behind attention-based architectures is attributed to their ability to capture *long-range interactions* among data tokens (such as words and image patches) via attention coefficients that are *global*, *learnable* and *adapted* to the input at test time. For example, while recurrent neural network architectures in natural language processing predict the next word in a sentence using information about a few previous words, self-attention mechanisms make a prediction based on interactions among all words. Similarly, while convolutional architectures in computer vision compute *local interactions* among image patches using weights that do not depend on the input image at test time, vision transformers compute global interactions that are adapted to the input at test time.

In this paper, we show that many of the key ideas behind attention, which we briefly summarize in Section 2, build upon a long history of prior work on manifold learning and image processing. In Section 3 we show that the **scaled dot product attention** mechanism is equivalent to kernel-based regression with the Gaussian kernel (Nadaraya, 1964; Watson, 1964), as recently pointed out in (Chaudhari et al., 2019; Zhang et al., 2021a), and that more general attention mechanisms can be obtained by choosing other kernels. We also show in Section 3 that the non-local means image denoising algorithm (Buades et al., 2005), which can also be understood as a form of kernel-based regression, is the main building block behind the vision transformer (ViT) (Dosovitskiy et al., 2020).

As a consequence, we argue that the key innovation of attention relative to kernel-based regression is not on its ability to capture global long-range interactions that are adapted to the input data (something that non-local means already does), but rather on the use of many learnable parameters for defining attention. In contrast, classical kernel methods typically tune only the kernel bandwidth.

In Section 4 we establish several connections between **masked attention** and Locally Linear Embedding (LLE) (Roweis & Saul, 2000; 2003). Specifically, we show that LLE learns a low-dimensional representation of a dataset using a *masked self-attention mechanism* where the masks are defined by the nearest neighbors of a data point. The resulting coefficients are not constrained to be nonnegative, thus allowing for both positive and negative attention. Moreover, they depend explicitly on multiple data tokens, unlike attention coefficients which depend only on a pair of tokens. We also show that LLE's training objective can be interpreted as a *fill in the blanks* self-supervised learning objective. However, a key limitation of LLE is that its local neighborhood is pre-specified, so a data point cannot attend to any other point. This issue is resolved by self-expressiveness, which connects every point to every other point and uses sparse regularization to reveal which points to attend to.

In Section 5 we show that **self-attention** is closely related the notion of self-expressiveness of Elhamifar & Vidal (2009; 2013); Vidal et al. (2016), wherein the data points to be clustered are expressed as linear combinations of other points with global coefficients that are adapted to the data and capture long-range interactions among data points. Such self-expressive coefficients are then used to define a data affinity matrix which is used to cluster the data. A first difference between self-attention and self-expressive coefficients is that the latter are not restricted to be non-negative, thus allowing for both positive and negative attention. A second difference is that self-expressive coefficients are not defined as a function of the tokens parametrized by learnable weights. Instead, the coefficients are learned directly using an unsupervised loss. A third difference is that self-expressive coefficients are typically regularized to be sparse or low-rank. As a consequence, we argue that the key innovation of self-attention relative to self-expressiveness is neither in its ability to capture global long-range interactions that are adapted to the data nor in the ability to learn such interactions (something that self-expressiveness already does), but rather on the fact that attention mechanisms use multiple attention-heads in parallel and are stacked into deep architectures.

We conclude with **future directions** on how to improve self-expressiveness using self-attention and vice versa. For example, we argue that the use of sparse regularization in (Elhamifar & Vidal, 2009; 2013) to automatically select the most relevant coefficients is a more principled way of handling a large number of tokens than restricting attention to arbitrary local neighborhoods, as done e.g., in criss-cross attention (Huang et al., 2019). To achieve this, we suggest unrolling the sparse encoding mechanism in order to induce sparse attention maps through multiple layers of attention. We conjecture this may not only improve self-attention-based architectures through the use of sparse regularizers on the attention coefficients, but also improve subspace clustering methods by using self-attention, as recently proposed in (Zhang et al., 2021b). This could also allow one to extend subspace clustering methods to nonlinear manifolds by stacking multiple layers of self-expressiveness.

## 2 TRANSFORMERS, ATTENTION AND SELF-ATTENTION

### 2.1 TRANSFORMER

The transformer architecture was originally designed for processing data sequences, e.g., a sequence of words in a sentence. As shown in Figure 1, each element of the sequence is first mapped to a vector space through a suitable embedding, e.g., a Word2vec embedding of a word. Since the architecture does not depend on the position and order of the input sequence, a *positional encoding* is added to the each input embedding. The resulting input *tokens* are then processed by a *multi-head attention layer*. This layer computes output tokens as linear combinations of input tokens weighted by *attention coefficients* designed to capture long-range interactions among input tokens, such as word associations. The output tokens are then processed by a residual connection followed by layer normalization, a feed-forward network such as an MLP, and another residual connection and normalization layer. Therefore, the main component of the transformer architecture is the (multi-head) attention layer, which we describe next.

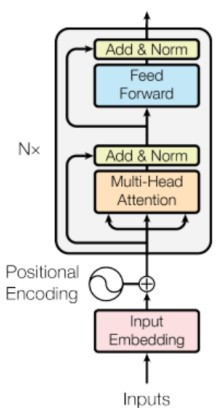

Figure 1: Transformer encoder architecture Vaswani et al. (2017).

## 2.2 ATTENTION

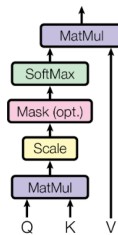

As illustrated in Figure 2, the *attention layer* is designed to capture long-range interactions among three types of input tokens: queries, keys and values. It does so by comparing *queries* to *keys* to produce a set of *attention coefficients*, which are then used to generate linear combinations of the *values*. Specifically, let us denote the queries by matrix $\boldsymbol{Q} = [\boldsymbol{q}_1, \ldots, \boldsymbol{q}_{N_q}] \in \mathbb{R}^{d \times N_q}$, the keys by matrix $\boldsymbol{K} = [\boldsymbol{k}_1, \ldots, \boldsymbol{k}_{N_k}] \in \mathbb{R}^{d \times N_k}$, and the corresponding values by matrix $\boldsymbol{V} = [\boldsymbol{v}_1, \ldots, \boldsymbol{v}_{N_k}] \in \mathbb{R}^{d_v \times N_k}$. The attention layer computes an *attention coefficient* $c_{ij} = \text{attn}(\boldsymbol{k}_i, \boldsymbol{q}_j) \in [0, 1]$ for each key-query pair and returns a linear combination of the values as follows:

Figure 2: Scaled dot product attention Vaswani et al. (2017).

$$\boldsymbol{z}_j = \sum_{i=1}^{N_k} \boldsymbol{v}_i c_{ij} \text{ or } \boldsymbol{Z} = \boldsymbol{V}\boldsymbol{C}, \text{ where } \boldsymbol{C} = \text{attn}(\boldsymbol{K}, \boldsymbol{Q}) \in [0, 1]^{N_k \times N_q}. \tag{1}$$

Intuitively, the attention coefficient $c_{ij}$ measures the importance of key $\boldsymbol{k}_i$ for representing query $\boldsymbol{q}_j$ and the representation $\boldsymbol{z}_j$ combines the values $\boldsymbol{v}_i$ that are most important for $\boldsymbol{q}_j$. The are many possible choices for the attention mechanism, including additive attention, multiplicative attention and dot product attention. A common choice is *scaled dot product attention*, which applies a softmax operator to the dot product of keys and queries scaled by the square root of their dimension, i.e.:

$$\boldsymbol{C} = \text{softmax}\left(\frac{\boldsymbol{K}^\top \boldsymbol{Q}}{\sqrt{d}}\right) = \frac{\exp(\boldsymbol{k}_i^\top \boldsymbol{q}_j / \sqrt{d})}{\sum_i \exp(\boldsymbol{k}_i^\top \boldsymbol{q}_j / \sqrt{d})}. \tag{2}$$

Since the coefficients are non-negative and add up to one, $\boldsymbol{z}_j$ is a convex combination of the values.

Let us illustrate the intuition behind attention using the following (overly simplified) examples:

1. Suppose we would like to translate sentences from French to English. Let $\boldsymbol{q}_j$ be a feature embedding for the $j$th word of a sentence in French, and let $\boldsymbol{k}_i = \boldsymbol{v}_i$ be an embedding for the $i$th word of the corresponding sentence in English. Ideally, the attention mechanism should be designed such that the coefficient $c_{ij}$ is large ($c_{ij} \approx 1$) only for key-query pairs $(i, j)$ that correspond to the translation of French word $i$ into English word $j$, in which case the output to French query $\boldsymbol{q}_j$ will be its translation into English $\boldsymbol{z}_j = \boldsymbol{v}_i$.

2. Suppose we are given an image-caption pair and we would like to find which regions in the image corresponds to each word in the caption. Assume we also have a collection of bounding boxes extracted from the image, e.g., using an object detector. Let the queries be feature embeddings for the words in the caption and the keys and values be CNN features extracted from the bounding boxes. Ideally, the attention mechanism should be designed such that $c_{ij}$ is large when the box $i$ corresponds to word $j$. That is, the attention mechanism is designed to tell us which regions to pay attention to for each word.

Of course, in order for multilingual word embeddings to align with each other, or for word embeddings to match image features, both features need to be mapped to a common latent space through a learnable transformation. We discuss such mappings in the next subsection in the context of self-attention, but such mapping also apply here.

## 2.3 SELF-ATTENTION

Let $\boldsymbol{X} = [\boldsymbol{x}_1, \ldots, \boldsymbol{x}_N] \in \mathbb{R}^{D \times N}$ denote a set of data tokens, such as words or image patches. The goal of self-attention is to capture long-range interactions among such tokens. Such interactions are captured by first transforming these tokens into keys, queries and values via learnable coefficient matrices $W_K \in \mathbb{R}^{d \times D}$, $W_Q \in \mathbb{R}^{d \times D}$, and $W_V \in \mathbb{R}^{d_v \times d}$, respectively, as follows:

$$\boldsymbol{K} = W_K \boldsymbol{X} \in \mathbb{R}^{d \times N}, \quad \boldsymbol{Q} = W_Q \boldsymbol{X} \in \mathbb{R}^{d \times N}, \quad \text{and} \quad \boldsymbol{V} = W_V \boldsymbol{X} \in \mathbb{R}^{d_v \times n}. \tag{3}$$

Then, we can define a set of transformed tokens using attention, e.g.:

$$\boldsymbol{Z} = \boldsymbol{V} \text{softmax}(\boldsymbol{K}^\top \boldsymbol{Q} / \sqrt{d}). \tag{4}$$

Let us illustrate the intuition behind self-attention using the vision transformer (ViT) proposed in (Dosovitskiy et al., 2020). As shown in Figure 2.3, the ViT divides an input image into a collection

of patches and maps those patches to a set of vectors via a learnable linear projection. Each projected patch is augmented with a positional encoding for the location of the patch in the image. Since ViT is designed for image classification, an additional (zero) token is added to the input of the transformer. This token is expected to capture class information and is learned during training. The transformer encoder processes all these tokens using self-attention. Specifically, new tokens are formed as linear combinations of patches weighted by attention coefficients that capture relationships among image patches. Moreover, attention coeffi-

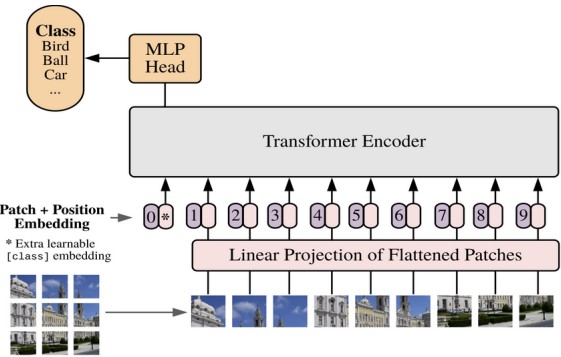

Figure 3: ViT architecture (Dosovitskiy et al., 2020).

cients relating the class token to patch tokens are expected to capture which patches to pay attention to in order to classify the image. The output class token is then passed through an MLP head to produce class probabilities. The network parameters (input class token, patch projection, self-attention weights, encoder MLP, MLP head) are learned using a cross-entropy loss for classification.

### 2.4 MASKED ATTENTION AND SPARSE ATTENTION

As discussed in the introduction, much of the success of attention-based architectures is attributed to the fact that attention layers capture *long-range interactions* among data tokens via attention coefficients that are *global* and *adapted* to the input data at test time. However, the use of the softmax operator often leads to dense attention maps, whose computation can be both memory and computationally intensive. Moreover, in applications such as document summarization, question answering or visual grounding, the attention maps are expected to be sparse. One approach to addressing this issue is to restrict non-zero attention coefficients to certain patterns, such as the criss-cross pattern proposed in (Huang et al., 2019). In the architecture shown in Figure 2, this is implemented via masks, hence the name *masked attention*. However, pre-defining local attention maps might miss important long-range interactions. As an alternative, Martins & Astudillo (2016) propose to substitute the softmax operator by a sparsemax operator, which directly induces sparse attention maps. However, it is not clear why doing so would automatically lead to selecting tokens that are more informative. This motivated He et al. (2021) to propose heuristics for combining attention maps in order to select informative tokens for fine-grained recognition. Overall, a rigorous method for inducing sparsity while maintaining the most informative long-range interactions remains elusive.

## 3 KERNEL REGRESSION, NON-LOCAL MEANS DENOISING AND ATTENTION

We begin with what arguably is one of the earliest incarnations of the idea of self-attention, namely kernel regression (Nadaraya, 1964; Watson, 1964). Interestingly, kernel regression is also at the root of a well-known image denoising algorithm, namely non-local means (Buades et al., 2005), which we show is strongly connected to the vision transformer (ViT) (Dosovitskiy et al., 2020).

### 3.1 KERNEL REGRESSION

The connection between attention and kernel regression was recently pointed out in Chaudhari et al. (2019); Zhang et al. (2021a). Kernel regression (Nadaraya, 1964; Watson, 1964) is a non-parametric method for fitting a function $f : \mathcal{X} \to \mathcal{Y}$ to samples $\{(\boldsymbol{x}_j, \boldsymbol{y}_j)\}_{j=1}^{N}$ drawn from $\mathcal{X} \times \mathcal{Y}$, which uses a kernel density estimator to approximate the minimum-mean-squared-error predictor $\hat{f}(\boldsymbol{x}) = \mathbb{E}(\boldsymbol{y}|\boldsymbol{x})$. Specifically, given a kernel $\kappa : \mathcal{X} \times \mathcal{X} \to \mathbb{R}$, Nadaraya (1964) and Watson (1964) show that one can estimate $\hat{f}(\boldsymbol{x})$ as a weighted combination of the values of $\boldsymbol{y}_j$, i.e.,

$$f(\boldsymbol{x}) = \sum_{j=1}^{N} \alpha(\boldsymbol{x}, \boldsymbol{x}_j)\boldsymbol{y}_j = \sum_{j=1}^{N} \frac{\kappa(\boldsymbol{x}, \boldsymbol{x}_j)}{\sum_{i=1}^{N} \kappa(\boldsymbol{x}, \boldsymbol{x}_i)} \boldsymbol{y}_j. \tag{5}$$

Intuitively, the weighting function $\alpha(\boldsymbol{x}, \boldsymbol{x}_j)$ encodes the relevance of $\boldsymbol{x}_j$ for predicting $f(\boldsymbol{x})$.

When $\kappa$ is a Gaussian kernel, $\kappa(\boldsymbol{x}, \boldsymbol{x}_j) = \exp(-\frac{\|\boldsymbol{x}-\boldsymbol{x}_j\|_2^2}{2\sigma^2})$, the expression in equation 5 in reduces to

$$f(\boldsymbol{x}) = \sum_{j=1}^{N} \frac{\exp(-\frac{\|\boldsymbol{x}-\boldsymbol{x}_j\|_2^2}{2\sigma^2})}{\sum_{i=1}^{N} \exp(-\frac{\|\boldsymbol{x}-\boldsymbol{x}_i\|_2^2}{2\sigma^2})} \boldsymbol{y}_j = \sum_{j=1}^{N} \mathrm{softmax}(\frac{-\|\boldsymbol{x}-\boldsymbol{x}_j\|_2^2}{2\sigma^2}) \boldsymbol{y}_j. \tag{6}$$

Therefore, **Nadayara-Watson regression is an attention mechanism** where the query is $\boldsymbol{q} = \boldsymbol{x}$, the keys are $\boldsymbol{k}_j = \boldsymbol{x}_j$, the values are $\boldsymbol{v}_j = \boldsymbol{y}_j$, and the attention function is softmax applied to minus the normalized squared distance between query and key. Further assuming that keys and queries are normalized as $\|\boldsymbol{x}\|_2 = \|\boldsymbol{x}_j\|_2 = 1$ so that $\|\boldsymbol{x} - \boldsymbol{x}_j\|_2^2 = 2(1 - \boldsymbol{x}^\top \boldsymbol{x}_j)$ yields scaled dot product attention:

$$f(\boldsymbol{x}) = \sum_{j=1}^{N} \frac{\exp(\frac{\boldsymbol{x}^\top \boldsymbol{x}_j}{\sigma^2})}{\sum_{i=1}^{N} \exp(\frac{\boldsymbol{x}^\top \boldsymbol{x}_i}{\sigma^2})} \boldsymbol{y}_j = \sum_{j=1}^{N} \mathrm{softmax}(\frac{\boldsymbol{x}^\top \boldsymbol{x}_j}{2\sigma^2}) \boldsymbol{y}_j. \tag{7}$$

Despite this obvious connection, we note that kernel regression with the Gaussian kernel is a local attention mechanism that is unable to capture general long-range interactions. This is because the attention weights depend upon the distance between the key and the query, which is adapted using only one tunable parameter: $\sigma$. When $\sigma$ is very small, although all pairwise interactions are computed, large interactions occur only in a local neighborhood, which results in a local attention mechanism. On the other hand, when $\sigma$ is very large all weights are similar and we get $f(\boldsymbol{x}) \approx \frac{1}{N} \sum \boldsymbol{y}_j$, which is clearly not an effective attention mechanism. Therefore, the key advantage of attention with respect to kernel regression is that it incorporates learnable linear transformations for both key and queries. Specifically, if we let $\boldsymbol{q} = W\boldsymbol{x}$ and $\boldsymbol{k}_j = W\boldsymbol{x}_j$, we obtain $\boldsymbol{q}^\top \boldsymbol{k}_j = \boldsymbol{x}^\top W^\top W \boldsymbol{x}_j$. In order for kernel regression to achieve such a learnable dot product, it would need to use a Gaussian kernel with a full covariance matrix $\Sigma$, and learn the resulting dot product which is given by $\boldsymbol{x}^\top \Sigma^{-1} \boldsymbol{x}_j$.

More generally, observe that the expression in equation 5 can be used to define new attention mechanisms by choosing different kernel function $\kappa$. For example, the Gaussian, Laplace and Wasserstein kernels are all members of the exponential family, as they are defined as the exponential of minus a squared distance, i.e., $\kappa(\boldsymbol{q}, \boldsymbol{k}) = \exp(-\mathrm{dist}(\boldsymbol{q}, \boldsymbol{k})^2)$. In this case, the resulting attention mechanism $\mathrm{attn}(\boldsymbol{q}, \boldsymbol{k}) = \mathrm{softmax}(-\mathrm{dist}(\boldsymbol{q}, \boldsymbol{k})^2)$ is defined based on a notion of similarity (Graves et al., 2014). On the other hand, it is not clear if all existing attention mechanisms (e.g., additive attention) can be written in terms of a kernel.

### 3.2 Non-local Means Denoising

As the name suggests, image denoising methods aim to remove noise in an mage. The most basic image denoising method is based on computing the average intensity of a set of neighboring pixels. Typically, a local Gaussian weighted average is used. Specifically, if $\boldsymbol{x}_j$ denotes the 2D coordinates of pixel $j$ and $\boldsymbol{y}_j$ denote its intensity or RGB values, the denoised image at pixel $\boldsymbol{x}$ takes the form in equation 5. Since $\sigma$ is typically chosen to be small (say 3-11 pixels) and Gaussian weights decay very quickly with the distance $\|\boldsymbol{x} - \boldsymbol{x}_j\|$, it is customary to restrict the sum in equation 5 to a neighborhood of $\boldsymbol{x}$ of size $\approx 3\sigma$. In this case, the sum becomes a convolution with a Gaussian filter. Therefore, **classical denoising is a local attention mechanism** with queries and keys denoting pixel locations (i.e., $\boldsymbol{q}_j = \boldsymbol{k}_j = \boldsymbol{x}_j$) and values denoting image intensities (i.e., $\boldsymbol{v}_j = \boldsymbol{y}_j$).

Non-local means introduces two key modifications to classical image denoising. First, it computes the weighted average of the intensities of all pixels, not just of a local neighborhood of $\boldsymbol{x}$, as in equation 5. Second, it uses a Gaussian kernel based on the intensity value $\boldsymbol{y}_j$ rather than the pixel location $\boldsymbol{x}_j$. This allows the algorithm to be non-local in that it finds other (possibly far away) pixels with similar intensities. Specifically, in its simplest form, non-local means denoises the image as

$$f(\boldsymbol{x}) = \sum_{j=1}^{N} \frac{\exp(-\frac{\|\boldsymbol{y}-\boldsymbol{y}_j\|^2}{2\sigma^2})}{\sum_{i=1}^{N} \exp(-\frac{\|\boldsymbol{y}-\boldsymbol{y}_i\|^2}{2\sigma^2})} \boldsymbol{y}_j. \tag{8}$$

Therefore, this simplified form of **non-local means denoising is a self-attention mechanism** with queries, keys and values denoting image brightness (i.e., $\boldsymbol{q}_j = \boldsymbol{k}_j = \boldsymbol{v}_j = \boldsymbol{y}_j$).

A slightly more general form of the non-local means algorithm computes a Gaussian kernel not on the intensities $\boldsymbol{y}$ and $\boldsymbol{y}_j$ of a single pixel, but on the intensities of patches centered at pixels $\boldsymbol{x}$ and $\boldsymbol{x}_j$, respectively. This allows the algorithm to attend to distant patches that are similar and hence

useful for denoising. Therefore, **non-local means denoising is an attention mechanism** where the queries and keys are the intensities of image patches and the values are the intensities of the central pixel. This connection had been noted in Wang et al. (2018), but surprisingly it is not mentioned in (Dosovitskiy et al., 2020). Indeed, notice that the steps of non-local means are equivalent to:

1. Extract a set of overlapping patches from the image.

2. Flatten these patches.

3. Apply an attention mechanism with the keys and queries being the flattened patches and the values being the intensity of their central pixel.

Therefore, **non-local means denoising is closely related to the vision transformer**, except that (a) there is no additional classification token, (b) the projection of patches is fixed as the identity rather than learned, (c) no positional encoding is added to the embedded patches, and (d) a single-head and single-layer self-attention mechanism is used without normalization or fully connected layers.

# 4 LOCALLY LINEAR EMBEDDING (LLE) AND LOCAL SELF-ATTENTION

In this section, we show that LLE learns a low-dimensional representation of a dataset by using a *masked local self-attention mechanism*. Specifically, we show that LLE coefficients can be interpreted as local attention weights with masks defined by the nearest neighbors. We note that LLE coefficients are not constrained to be nonnegative, thus allowing for both positive and negative attention, and LLE coefficients depend explicitly on multiple data tokens, unlike additive attention and scaled dot product which depend only on a pair of tokens (except for softmax). Finally, we show that LLE's training objective can be interpreted as a *fill in the blanks* self-supervised learning objective.

## 4.1 LOCALLY LINEAR EMBEDDING

Let us first recall that LLE aims to learn a locally-linear low-dimensional embedding $\{\boldsymbol{y}_j\}_{j=1}^N \subset \mathbb{R}^d$ of a given data set $\{\boldsymbol{x}_j\}_{j=1}^N \subset \mathbb{R}^D$, where $D$ is the data dimension and $d \ll D$ is the embedding dimension. LLE computes this low-dimensional embedding by first expressing each data point $\boldsymbol{x}_j$ as an affine combination of its $K$-nearest neighbors, i.e., by finding coefficients $c_{ij} \in \mathbb{R}$ such that $\boldsymbol{x}_j \approx \sum_{i \in N_j} \boldsymbol{x}_i c_{ij}$ and $\sum_{i \in N_j} c_{ij} = 1$, where $N_j \subset \{1, \ldots, N\}$ is the set of $K$-nearest neighbors of $\boldsymbol{x}_j$. More specifically, LLE finds the coefficients by minimizing the reconstruction error

$$\min_{\{c_{ij}\}} \sum_{j=1}^N \left\| \boldsymbol{x}_j - \sum_{i \in N_j} \boldsymbol{x}_i c_{ij} \right\|_2^2 \quad \text{s.t.} \quad \sum_{i \in N_j} c_{ij} = 1 \quad \forall j = 1, \ldots, N. \tag{9}$$

Once these coefficients have been found, LLE finds a low-dimensional representation that is centered at the origin, has unit covariance, and minimizes the same reconstruction error, i.e.

$$\min_{\{\boldsymbol{y}_j\}} \sum_{j=1}^N \left\| \boldsymbol{y}_j - \sum_{i \in N_j} \boldsymbol{y}_i c_{ij} \right\|_2^2 \quad \text{s.t.} \quad \sum_{j=1}^N \boldsymbol{y}_j = 0 \quad \text{and} \quad \sum_{j=1}^N \boldsymbol{y}_j \boldsymbol{y}_j^\top = I. \tag{10}$$

## 4.2 LLE VERSUS LOCAL-ATTENTION

In order to show that LLE uses a masked local self-attention mechanism, observe that the coefficient $c_{ij}$ in the expression $\boldsymbol{x}_j \approx \sum_{i \in N_j} \boldsymbol{x}_i c_{ij}$ can be interpreted as an *attention weight* that measures the contribution of point $\boldsymbol{x}_i$ to point $\boldsymbol{x}_j$. Specifically, note that the optimization problem in equation 9 can be decoupled as $N$ optimization problems, one for each $\boldsymbol{x}_j$, and that the optimal coefficients for $\boldsymbol{x}_j$ are a function of the *query* $\boldsymbol{x}_j$ and the *keys* $\{\boldsymbol{x}_i\}_{i \in N_j}$, i.e., $\{c_{ij}^*\}_{i \in N_j} = f(\boldsymbol{x}_j, \{\boldsymbol{x}_i\}_{i \in N_j})$[1]. All other coefficients $\{c_{ij}^*\}_{i \notin N_j}$ are set to zero, thus the $K$ nearest neighbors define a local attention

---

[1] If $j_1, j_2, \ldots, j_K$ are the indices of the $K$-NN of $\boldsymbol{x}_j$, $\boldsymbol{c}_j = [c_{j_1,j}, c_{j_2,j}, \ldots, c_{j_K,j}]^\top \in \mathbb{R}^K$ is its vector of affine coefficients and $G_j = [g_{il}^j] \in \mathbb{R}^{K \times K}$ is its local Gram matrix defined as $g_{il}^j = (\boldsymbol{x}_i - \boldsymbol{x}_j)^\top (\boldsymbol{x}_l - \boldsymbol{x}_j)$ if $\boldsymbol{x}_i$ and $\boldsymbol{x}_j$ that are $K$-NN of $\boldsymbol{x}_j$, then the optimal solution is $\boldsymbol{c}_j = \frac{G_j^{-1} \mathbf{1}}{\mathbf{1}^\top G_j^{-1} \mathbf{1}}$.

mask. Note also that the constraint in equation 9 ensures the weights add up to 1 without requiring a softmax post-processing. Finally, notice that the training objective in equation 9 is a fill in the blanks objective, where the nearest neighbors of $\boldsymbol{x}_j$, $\{\boldsymbol{x}_i\}_{i \in N_j}$, are used to predict the missing token $\boldsymbol{x}_j$.

Despite these similarities between LLE and existing attention mechanisms, there are some important differences. First, most existing attention mechanisms compute a score function applied to a single query-key pair and then apply the softmax function so that attention weights are between 0 and 1. In contrast, LLE coefficients are not constrained to be nonnegative, thus allowing for both positive and negative attention. Moreover, LLE coefficients depend on both the query and multiple keys. Another difference, perhaps the most important one, is that in most existing attention mechanisms $c_{ij}$ is a parametrized function of the query-key pair whose weights are learned during training. In sharp contrast, LLE learns the values of $c_{ij}$ directly, which makes it more difficult to evaluate the coefficients for new data, as the optimization problem in equation 9 needs to be solved anew.

Despite these differences, we note many of the key ingredients of attention (key, query, value, mask) were already present in the original LLE formulation, albeit for different purposes. In particular, LLE is based on the idea that each query attends its $K$ nearest neighbors by writing itself as an affine combination of such neighbors. The attention weights thus capture the local geometry of the data manifold and are hence used to find the low-dimensional embedding as per equation 10.

## 5 SUBSPACE CLUSTERING, SELF-EXPRESSIVENESS AND SELF-ATTENTION

A key limitation of LLE is that its local neighborhood is pre-specified, so a data point cannot attend to any other point. In this section we show that this issue is resolved by self-expressiveness (El-hamifar & Vidal, 2009; 2013; Vidal et al., 2016), which connects every point to every other point and uses sparse regularization to reveal which points to attend to. Specifically, we show that self-expressiveness based subspace clustering methods such as *sparse subspace clustering* (Elhamifar & Vidal, 2009; 2013; Wang & Xu, 2013), *low-rank subspace clustering* (Liu et al., 2010; Vidal & Favaro, 2014), least squares regression (Lu et al., 2012) and extensions (Wang et al., 2013) compute a data affinity using a *global masked self-attention mechanism* where the queries, keys and values are the data points to be clustered, and the self-expressive coefficients of a data point are designed to *attend* to other points in the same subspace. We note, however, that self-expressive coefficients are not constrained to be nonnegative, thus allowing for both positive and negative attention. We also show that self-expressive coefficients are *global* in that they truly depend on multiple data points, unlike most attention mechanisms that depend only on a pair of tokens (except for softmax). Finally, we show that the subspace clustering training objective can be interpreted as a *fill in the blanks* self-supervised learning objective where each data point is regressed with respect to all other data points.

### 5.1 SUBSPACE CLUSTERING AND SELF-EXPRESSIVENESS

Subspace clustering refers to the problem of clustering data drawn from a union of subspaces. Self-expressiveness based methods solve this problem by expressing each data point as a linear combination of all other data points. The resulting self-expressive coefficients reveal information about which points belong to the same subspace, hence they can be used to define a suitable data affinity matrix. The clustering of the data is then obtained by applying spectral clustering to such an affinity.

More formally, let $\boldsymbol{X} = [\boldsymbol{x}_1, \ldots, \boldsymbol{x}_N]$ be a set of points drawn from a union of $n$ subspaces of $\mathbb{R}^D$ of dimension $d \ll D$ which we wish to cluster. Assume that the data from each subspace is sufficiently rich so that any $d$ points from one group span the subspace associated to that group. Then, each data point $\boldsymbol{x}_j$ can be expressed as a linear combination of $d$ other points in its own subspace. That is, for all $j = 1, \ldots, N$, there exist at most $d$ non-zero coefficients $c_{ij} \in \mathbb{R}$ such that:

$$\boldsymbol{x}_j = \sum_{i \neq j} \boldsymbol{x}_i c_{ij}, \quad \text{or} \quad \boldsymbol{X} = \boldsymbol{X}\boldsymbol{C} \ \text{and} \ \text{diag}(\boldsymbol{C}) = 0, \tag{11}$$

where $\boldsymbol{C} \in \mathbb{R}^{N \times N}$ is the matrix of coefficients. Notice that in the above constraint data points are expressed as linear combinations of each other, hence the name *self-expressiveness*.

Since our goal is to use the self-expressive coefficients to define an affinity matrix for clustering the data, ideally the coefficients should have the property that $c_{ij} \neq 0$ only if points $\boldsymbol{x}_i$ and $\boldsymbol{x}_j$ are in

the same subspace. Coefficients that satisfy such a property are guaranteed to exist since a point can always be expressed in terms of $d$ points in its own subspace. Moreover, if $d \ll N$, i.e., if the subspaces are low-dimensional and the number of data points is sufficiently large, such coefficients are *sparse*. This motivates the sparse subspace clustering objective (Elhamifar & Vidal, 2009)

$$\min_{\{c_{ij}\}} \|\boldsymbol{x}_j - \sum_{i \neq j} \boldsymbol{x}_i c_{ij}\|_2^2 + \lambda \sum_{i \neq j} |c_{ij}|, \quad \text{or} \quad \min_{\boldsymbol{C}:\text{diag}(\boldsymbol{C})=0} \|\boldsymbol{X} - \boldsymbol{X}\boldsymbol{C}\|_F^2 + \lambda \|\boldsymbol{C}\|_1, \quad (12)$$

where the first term measures how well a data point is reconstructed in terms of other data points, the second term uses $\ell_1$ regularization to encourage sparsity, and $\lambda > 0$ is a regularization parameter. More generally, one can use other regularizers $\Theta$ and write the objective in terms of the matrix $\boldsymbol{C}$

$$\min_{\boldsymbol{C}} \|\boldsymbol{X} - \boldsymbol{X}\boldsymbol{C}\|_F^2 + \lambda \Theta(\boldsymbol{C}). \quad (13)$$

Once the coefficients have been computed (see next subsection), it is common to select the largest nonzero coefficients to induce additional sparsity and to normalize the columns of $\boldsymbol{C}$ so that they add up to one (Elhamifar & Vidal, 2013). Alternatively, once can add an $\ell_1$-normalization constraints to the optimization problem in equation 13, as is commonly done in affine subspace clustering (Elhamifar & Vidal, 2013; Li et al., 2018; You et al., 2019). Interestingly, it appears that sofmax normalization of the coefficients has never used in the subspace clustering literature. Finally, given $\boldsymbol{C}$, the data is clustered by applying spectral clustering to an affinity matrix that is often constructed by symmetrizing the absolute values of the self-expressive coefficients, i.e., $\boldsymbol{A} = |\boldsymbol{C}| + |\boldsymbol{C}^\top|$.

## 5.2 SELF-EXPRESSIVE COEFFICIENTS FOR DIFFERENT REGULARIZERS

The least squares regression approach (Lu et al., 2012) uses $\Theta(\boldsymbol{C}) = \|\boldsymbol{C}\|_F^2$ and gives a closed form solution for $\boldsymbol{C} = (\boldsymbol{X}^\top \boldsymbol{X} + \lambda I)^{-1} \boldsymbol{X}^\top \boldsymbol{X} = \boldsymbol{V}(\Sigma^2 + \lambda I)^{-1}\Sigma^2 \boldsymbol{V}^\top$, where $\boldsymbol{X} = \boldsymbol{U}\Sigma\boldsymbol{V}^\top$ is the SVD of the data. Therefore, the self-expressive coefficient $c_{ij} = \boldsymbol{v}_i^\top (\Sigma^2 + \lambda I)^{-1}\Sigma^2 \boldsymbol{v}_j$ is a weighted dot product of rows of $\boldsymbol{V}$. When $\lambda$ is large enough we get a scaled dot product of the data points

$$\boldsymbol{C} \approx \frac{1}{\lambda} \boldsymbol{X}^\top \boldsymbol{X}. \quad (14)$$

The low-rank subspace clustering approach (Liu et al., 2010; Vidal & Favaro, 2014) uses a nuclear norm regularizer $\Theta(\boldsymbol{C}) = \|\boldsymbol{C}\|_*$ to induce low-rank coefficients. The solution can be computed in closed form from the SVD of the data as $\boldsymbol{C} = \boldsymbol{V}\text{ReLU}_\lambda(\Sigma)\boldsymbol{V}^\top$, where $\text{ReLU}_\lambda(x) = \max(x - \lambda, 0)$. As before, this can be interpreted as a weighted dot product of rows of $\boldsymbol{V}$, except that some weights can be zero to induce low-rank.

The sparse subspace clustering approach (Elhamifar & Vidal, 2009; 2013; Wang & Xu, 2013) uses the $\ell_1$ norm $\Theta(\boldsymbol{C}) = \|\boldsymbol{C}\|_1$ to induce sparse coefficients. In this case, the coefficients cannot be computed in closed form. However, a common approach is to use the Iterative Shrinkage Thresholding Algorithm (ISTA) proposed by (Beck & Teboulle, 2009), which can be written as:[2]

$$\boldsymbol{C}_{k+1} = \text{ReLU}_\lambda\big((I - \epsilon \boldsymbol{X}^\top \boldsymbol{X})\boldsymbol{C}_k + \epsilon \boldsymbol{X}^\top \boldsymbol{X}\big) = \text{ReLU}_\lambda\big(\boldsymbol{C}_k + \epsilon \boldsymbol{X}^\top (\boldsymbol{X} - \boldsymbol{X}\boldsymbol{C}_k)\big), \quad (15)$$

where $\epsilon > 0$ is a step size. We note that equation 15 is the point of the departure for the unrolling approach proposed in (Gregor & LeCun, 2010), which connects sparse coding with neural networks. In that approach, the iterates are interpreted as activation functions of a neural network and the linear transformations $(I - \epsilon \boldsymbol{X}^\top \boldsymbol{X})$ and $\boldsymbol{X}^\top \boldsymbol{X}$ as learnable weights.

As a future research direction, we suggest further exploring the connections between sparse subspace clustering and transformers via unrolling, which we conjecture will allow us to extend subspace clustering to nonlinear manifolds through the use of (deep) multi-layer attention models. More specifically, notice that we can partially re-interpret equation 15 as the update of one attention layer. This is because in equation 15, the term $\boldsymbol{X} - \boldsymbol{X}\boldsymbol{C}_k$ can be interpreted as applying attention $\boldsymbol{C}_k$ to input data $\boldsymbol{X}$ and then adding a (negative) residual connection with the input $\boldsymbol{X}$. Then, the multiplication by $\boldsymbol{X}^\top$ in in equation 15 and the ReLU nonlinearity can be interpreted as the feedforward layer of the transformer. Of course, the analogy is not perfect because the addition of $\boldsymbol{C}_k$ is not quite a residual connection.

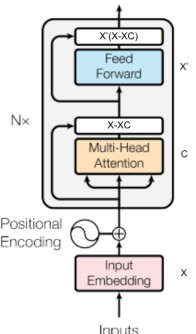

Figure 4: Towards a sparse transformer?

---

[2]We have neglected the constraint $\text{diag}(\boldsymbol{C}) = 0$ for ease of exposition

### 5.3 SELF-EXPRESSIVENESS VERSUS SELF-ATTENTION

Notice from equation 11 that self-expressiveness can be interpreted as a self-attention mechanism where the query $q_j = x_j$ is expressed as a linear combination of all values $v_i = x_i, i = 1, \ldots, N$, with attention coefficients $c_{ij}$ determined by the queries $q_j = x_j$ and the keys $k_i = x_i$. However, we note that self-expressive coefficients (SEC) are more general than self-attention coefficients.

1. SEC are not restricted to be nonnegative, allowing for both positive and negative attention.

2. SEC are not restricted to be an explicit function of a single key-query pair. For example, the closed form solution to least squares regression has a term of the form $(\lambda I + X^\top X)^{-1}$, which makes $c_{ij}$ a function of all key-query pairs. The only case where self-expressive coefficients yield an explicit function of a single key-query pair is when $\lambda$ is large that as per equation 14, which resembles a scaled dot product attention.

3. SEC are not defined as a function of the tokens parametrized by learnable weights. Instead, the coefficients are learned directly using an unsupervised loss. This is, however, a potential disadvantage of self-expressiveness, as it makes it difficult to compute coefficients at test time. This issue is addressed in (Zhang et al., 2021b) by using learnable coefficients.

4. SEC are typically regularized to be sparse or low-rank. We argue that the use of sparse regularization in (Elhamifar & Vidal, 2009; 2013) to automatically select the most relevant coefficients is a more principled way of handling a large number of tokens than restricting attention to arbitrary local neighborhoods, e.g., in criss-cross attention (Huang et al., 2019).

As a consequence, we argue that the key innovation of self-attention relative to self-expressiveness is neither in its ability to capture global long-range interactions that are adapted to the data nor in the ability to learn such interactions (something that self-expressiveness already does), but rather on the fact that attention mechanisms have been stacked into deep architectures and with multiple attention-heads in parallel. As suggested in the previous section, further exploring the connections between sparse subspace clustering and transformers via unrolling might lead to (deep) multi-layer subspace clustering models. Alternatively, one may use attention mechanisms to parametrize self-expressive coefficients, as recently suggested in (Zhang et al., 2021b).

### 5.4 SPARSE CODING AND SPARSE ATTENTION

The connections made between self-expressiveness and self-attention also suggest new directions towards improving transformers via sparse encoding. Specifically, recall that in standard sparse coding, a data point $y$ is expressed as a sparse linear combination of dictionary atoms $A = [a_1, \ldots, a_N]$ with coefficients $c$ by solving the optimization problem

$$\min_c \|y - Ac\|_2^2 + \lambda\|c\|_1. \tag{16}$$

Reinterpreting the data point $y$ as the query $q$ and the dictionary $A$ as the set of keys $K$, and solving the problem for multiple queries $Q$ leads to an attention mechanism of the form

$$Z = VC^* \quad \text{where} \quad C^* = \arg\min_C \|Q - KC\|_2^2 + \lambda\|C\|_1. \tag{17}$$

Since solving a sparse coding problem can be costly, we unroll sparse coding iterates and obtain:

$$Z = VC_K \quad \text{where} \quad C_{k+1} = \text{ReLU}_\lambda((I - \epsilon K^\top K)C_k + \epsilon K^\top Q), \quad k = 1, \ldots, K. \tag{18}$$

Observe that the update has a rather interesting structure. The term $K^\top K$ is dot product self-attention, while the term $K^\top Q$ is dot product attention. Therefore, the update equation combines standard attention and self-attention to produce a new sparse attention map.

## 6 CONCLUSIONS

We have shown that attention builds upon a long history of prior work on manifold learning and image processing, including methods such as kernel-based regression, non-local means, locally linear embedding, subspace clustering and sparse coding. In particular, we showed that many of the key ideas behind attention, such as its ability to capture global long-range interactions that are learned and adapted to the input, had already appeared in the literature. Therefore, the key innovations of attention mechanisms relative to prior art are the use of many learnable parameters, and multiple heads and layers.

**Ethics Statement**   This work focuses on understanding the principles behind transformers and connecting it to well established topics in machine learning research. The research conducted in the framework of this work raises no ethical issues or any violations vis-a-vis the ICLR Code of Ethics.

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
