# OpenReview forum: "Attention: Self-Expression Is All You Need"
_ICLR.cc/2022/Conference — ICLR 2022 Submitted_

### Official Review · Reviewer_vLjt · 2021-10-20

**Correctness:** 3
**Technical Novelty And Significance:** 2
**Empirical Novelty And Significance:** 2
**Recommendation:** 5
**Confidence:** 4

**Main Review:**

This is a review paper to discusses the relationships between the attentions and the traditional (coding) methods, including attention and kernel-based regression, masked attention and locally linear embeding, self-attention and self-expressiveness.

ves:
+ The problem of the principles behind attention seems interesting.
+ This paper is well-written and is easy to follow.
+ The authors show some future directions on how to improve self-expressiveness using self-attention and vice versa.

Concerns:
- The title “attention: self-expression is all you need” seems too over-expression due to the following reason. The self-expression is that “each data point in a union of subspaces can be efficiently reconstructed by a combination of other points in the dataset.” (Elhamifar & Vidal 2009; 2013) This leads to the reconstruction loss in the Eq. (12). However, self-attention is to use itself to choose itself, that is, the calculation formula is defined in the Eq. (4), and it’s loss depends on the different tasks. It is clear that the self-attention does not need the self-expression or the self-expression is one case of the self-attention.
- This paper discuses the three parts. Could you propose a unified form to cover all the three parts?
- In the subsection 5.3, the authors note that self-expressive coefficients (SEC) are more general than self-attention coefficients. However, SEC still have some shortcomings.
1) SEC must result in over-high time and space complexities because it considers self-expressiveness. Thus, the current SEC methods cannot apply into large-scale datasets.
2) Although SEC are typically regularized to be sparse or low-rank, it is well-known that sparse or low-rank regularization lead to expensive costs.
- In the subsection 5.4, the authors consider the Eq. (17) instead of the Eq. (4). I have two suggestions.
1) Could you prove the equivalence between the Eq. (17,18) and (4)?
2) Could you provide some rough experimental results to verify the effectiveness of the Eq. (18)?


**Summary Of The Paper:**

This review paper tries to study the principles behind attention and its connections with prior art using the self-expression methods (e.g., kernel-based regression, non-local means, locally linear embedding, subspace clustering).

**Summary Of The Review:**

I recommend this score since this paper is to consider an interesting problem, some suggestions are hard to implement in deep models on large-scale datasets (millions), and there are no experimental results to support the arguments.

---

### Official Review · Reviewer_K4hc · 2021-11-03

**Correctness:** 3
**Technical Novelty And Significance:** 3
**Empirical Novelty And Significance:** Not applicable
**Recommendation:** 5
**Confidence:** 3

**Main Review:**

Strengths:

+    The paper does a literated survey of the prior work that has connection to self-attention. It offers various perspectives on the self-attention module, and it helps us slowing down our steps and really think about what's the real magic in self-attention and how can we improve it by taking inspirations from other problems and areas.
+    The proposed directions for future work are quite interesting, including using self-attention for manifold clustering, and connecting sparse reconstruction with self-attention.
+    The paper is easy to follow.

Weaknesses:

The main concern for me is the contribution of this paper. The main body of the paper is dedicated to showing the relation between self-attention and prior arts.
-    First of all, the relation between self-attention and kernel regression and non-local mean denoising has already pointed out by previous work, as cited by the authors.
-    Second of all, the relation between self-attention and locally linear embedding and self-expression in subspace clustering seems not so close to me. For example, self-expression estimates the affinity of the points sampled from the same subspace. But could we assume that different words in a sentence are sampled from a low-dimensional subspace? Despite the similar form (weighted average of data points), self-attention and the prior arts are targeting at different problems, so I think it's not convincing to say that self-attention is somehow "reinvented" from previous algorithms.
-    The paper proposes several conjectures on how self-attention could be improved or be applied to other problems. However, the authors didn't take a step further into any of them. The contribution would be much bigger if the authors could dive deep into one of the ideas to potentially improve self-attention or apply it in other problems. For example, the authors show that self-attention is related to non-local means and also to low-rank subspace clustering. It's well known that non-local means and low-rank subspace clustering both have the effect of removing certain noises in the data. Is it a possible way to explain that transformers are more robust than CNNs, as stated in recent works [1]?

[1] Paul S, Chen P Y. Vision transformers are robust learners[J]. arXiv preprint arXiv:2105.07581, 2021.


**Summary Of The Paper:**

This paper surveys several lines of prior work which has connection to the self-attention module in transformers. Specifically, the authors show that self-attention has the similar form with the kernel regression and non-local mean algorithm. They also demonstrate that locally linear embedding and self-expression algorithm for subspace clustering have the form of representing one data point by weighted sum of other data points, and thus are connected to the weighted sum of values in self-attention. Based on these observations, the authors argue that the innovation of self-attention is not modeling the long-range relation, which is also proposed in prior work, but the learnable parameters and the multi-head design. The authors also suggest several directions for future work, such as using self-attention for manifold clustering.

**Summary Of The Review:**

The paper explains the relation between self-attention and several previous algorithms and its implications, offering different perspectives on the self-attention module. However, the contribution is lacking in both justifying if self-attention and prior arts are functionally similar or just similar in the form, and exploring how the relations explained in the paper can help us better design self-attention or apply it in other areas.

---

### Official Review · Reviewer_Upbz · 2021-11-05

**Correctness:** 3
**Technical Novelty And Significance:** 3
**Empirical Novelty And Significance:** 3
**Recommendation:** 5
**Confidence:** 4

**Main Review:**

In general, this is an interesting paper with good insights. It provides holistic views towards one of the central parts of deep learning, attention, through many excellent learning methods developed before. However, I would not like to recommend acceptance because of the lack of theoretical and practical evidence to support and verify some basic claims as well as substantially ignore the progress within the deep learning community to bridge the attention method with prior arts listed in this paper.

It would be a successful work for the ICLR blog track but not the ICLR main conference. Many ideas are clearly of merit to inspire further research but not formal and supported enough as scientific work. I sincerely recommend submitting this work to the ICLR blog track this year, if the authors can sufficiently revise the writing style to suit a blog.

## Strengths

1. The paper is well-written, especially for explaining the key ideas of previous works. The intuition behind the connection between attention and prior arts is clear.
2. The insights provided by this paper are valuable even though not novel enough. Especially for many researchers starting their careers after the booming of deep learning, the discussion sketches a holistic and detailed perspective towards the essence of many previous works and how they are related to attention. This will be of interest to the community.

## Weaknesses

1. It is hard to define the scientific value of this paper because it fails to provide theoretical claims and misses experimental results to verify its implication based on self-expressiveness, i.e., the sparse attention mechanism inspired by the optimization of sparse coding. A solid work requires a systematically organized logic line supported by theorems or precise experiments to verify some ideas. Also, integrating these principled methods into neural networks is not such easy, especially due to the curse of training issues, computation budget, etc.
2. Although this work itself sufficiently discusses the correlation between attention and classic methods and clearly explains why these classic methods are related to attention or vice versa, it ignores many related works in the community that have essentially similar ideas or contributions. A simple list is as follows.

   For bridging attention with classic methods,
   - [1] Lu, J., Yao, J., Zhang, J., Zhu, X., Xu, H., Gao, W., Xu, C., Xiang, T., & Zhang, L. (2021). SOFT: Softmax-free Transformer with Linear Complexity. ArXiv, abs/2110.11945.
   - [2] Xiong, Y., Zeng, Z., Chakraborty, R., Tan, M., Fung, G.M., Li, Y., & Singh, V. (2021). Nyströmformer: A Nyström-Based Algorithm for Approximating Self-Attention. AAAI.
   - [3] Geng, Z., Guo, M., Chen, H., Li, X., Wei, K., & Lin, Z. (2021). Is Attention Better Than Matrix Decomposition? ICLR.
   - [4] Li, X., Zhong, Z., Wu, J., Yang, Y., Lin, Z., & Liu, H. (2019). Expectation-Maximization Attention Networks for Semantic Segmentation. ICCV.

   The framework by Hamburger[3] is strongly correlated with the self-expressiveness in this work.

   Considering the close connection between attention and Graph neural networks, here is a list of GNN papers inspired by classic methods.
   - [5] Yang, Y., Liu, T., Wang, Y., Zhou, J., Gan, Q., Wei, Z., Zhang, Z., Huang, Z., & Wipf, D.P. (2021). Graph Neural Networks Inspired by Classical Iterative Algorithms. ICML.
   - [6] Liu, X., Jin, W., Ma, Y., Li, Y., Liu, H., Wang, Y., Yan, M., & Tang, J. (2021). Elastic Graph Neural Networks. ICML.
   - [7] Zhu, M., Wang, X., Shi, C., Ji, H., & Cui, P. (2021). Interpreting and Unifying Graph Neural Networks with An Optimization Framework. Proceedings of the Web Conference 2021.
   - [8] Huang, Q., He, H., Singh, A., Lim, S., & Benson, A.R. (2021). Combining Label Propagation and Simple Models Out-performs Graph Neural Networks. ICLR.
   - [9] Wang, H., & Leskovec, J. (2020). Unifying Graph Convolutional Neural Networks and Label Propagation. ArXiv, abs/2002.06755.
   - [10] Wilder, B., Ewing, E., Dilkina, B.N., & Tambe, M. (2019). End to end learning and optimization on graphs. NeurIPS.

   Optimization-inspired methods [5,6,7] are strongly related as well.

**Summary Of The Paper:**

This paper studies the correlation between the attention mechanism and many prior arts. Heuristically, this paper links the currently hot topic, attention mechanism, with many milestones works in the past decades before the deep learning era, including subspace learning, sparse coding, kernel regression, non-local means, etc. Among these classic methods, many can be formulated as optimization problems. The minimizer to the optimization problem can be treated as the output of attention modules, and the optimization algorithm can be understood as the ``architecture'' of attention. Specifically, this work highlights the potential of sparse coding in designing the sparse attention mechanism when considering the self-expressiveness in subspace learning is essentially profoundly connected to the formulation of self-attention.

**Summary Of The Review:**

It would be a successful work for the ICLR blog track but not the ICLR main conference. Many ideas are clearly of merit to inspire further research but not formal and supported enough as scientific work. I sincerely recommend submitting this work to the ICLR blog track this year, if the authors can sufficiently revise the writing style to suit a blog.

---

### Public Comment · ~Xinyu_Chen5 · 2024-03-21
**Rethink Equation (15) for Iterative Shrinkage Thresholding Algorithm**

Thank you for this fascinating work for connecting transformers with the prior art! I hope to leave two comments on Eq. (15) in the manuscript.

- If this equation follows the proximal gradient descent, the threshold should be $\lambda\cdot\epsilon$, instead of $\lambda$.

- The second concern is the relationship between ReLU and proximal operator. To the best of my knowledge, they are equivalent only if $x\geq-\lambda$ in

\begin{equation}
\operatorname{ReLU}_{\lambda}(x) = \max(x−\lambda, 0),
\end{equation}

because the proximal operator of $\ell_1$-norm is

\begin{equation}
\mathcal{S}_{\lambda}(x)=\operatorname{sign}(x)\cdot\max(|x|-\lambda,0).
\end{equation}

---

### Decision · Program_Chairs · 2022-01-20

**Decision:**

Reject

**Comment:**

This paper points out connections between the self-attention module in transformers and some prior art, including kernel regression, the non-local mean algorithm, locally linear embeddings, and the self-expression algorithm for subspace clustering. Based on these observations, the authors argue that the innovation of self-attention is not modeling the long-range relation, which is also proposed in prior work, but the learnable parameters and the multi-head design. The authors also suggest several directions for future work, such as using self-attention for manifold clustering.

Reviewers pointed out several weaknesses with this paper: that some connections (e.g. connection to kernel regression) had been pointed out before, that the relation between self-attention and locally linear embedding and self-expression in subspace clustering is a bit nuanced, as pointed out by one of the reviewers, and that while some speculative future directions might be interesting, the paper falls short in actually trying some of them out empirically, or building a proof-of-concept.

In the discussion period, the authors pointed out that this is a position paper (which unfortunately was not expressed so assertively in their submission), which according to their view liberates them from digging deeper and test empirically some of these connections and speculative directions. According to the authors, a core contribution of their position paper is that "it expresses the opinion that the original attention paper failed to cite and acknowledge that attention mechanisms build upon a series of prior works in sparse coding, subspace clustering, and locally linear embedding."

There are no specific guidelines to review position papers at ICLR that I know of, but I will base my assessment on the assumption that a good position paper should:
- provide a good historical perspective of a subject
- connect previously unrelated lines of work in non-obvious ways
- inspire the research community to look at new directions.

While a good position paper can be extremely valuable and enlightening, I am not convinced that this particular paper achieves either of the goals above, and therefore it is my opinion that it does not deserve publication at ICLR.

As pointed out both by the authors and the reviewers, the connection between self-attention and kernel regression and non-local mean denoising is not new, and so it is not an original contribution of this paper. The relation between self-attention and locally linear embedding and self-expression in subspace clustering appears to be new, but this relation is a bit nuanced, as pointed out by one of the reviewers.

The tone of this position paper is that some of these connections were missing in the original attention paper -- the authors say "attention did not properly acknowledge prior art" in one of their responses (it is not clear if they are referring to Bahdanau et al.'s attention paper or to Vaswani et al.'s transformer paper). However, the historical perspective of how attention mechanisms came to be seems to be missing from this position paper -- attention has been proposed by Bahdanau et al. for machine translation, inspired by the idea of word alignment that has been prevalent in machine translation for decades. Later, in the transformer paper, self-attention was suggested as an alternative to recurrent and convolutional models for machine translation (note that self-attention has been used before the transformer paper, see e.g. [1]). While a theoretical connection with kernel regression etc. exists, this was not related to the original motivation of these works. There are many ways of arriving at the same construction! And given the simplicity of attention mechanisms it doesn't surprise me that connections with other lines of research exist. Had they been noticed, they would probably be a parenthesis in the original papers, because attention is derived there in a much more direct way (this doesn't mean that the connections aren't interesting, but that they are not _essencial_ to the construction).

In their response, the authors dismissed a constructive suggestion from one of the reviewers which in my opinion would have strengthen this paper -- the connection with graph neural networks. If the point of the paper is to point out past research that connects fundamentally to the idea of attention mechanisms, why leaving this out?

In sum, in my view this paper lacks the rigor, the insight, and the historical perspective that should characterize a strong position paper, and as such I cannot recommend acceptance. I strongly suggest that the authors take into account some of the insightful suggestions given by the reviewers in future iterations of their work.

[1] Ankur Parikh, Oscar Täckström, Dipanjan Das, and Jakob Uszkoreit. A decomposable attention model. In Empirical Methods in Natural Language Processing, 2016.